



# Seeking TOA SW Flux Closure over Synthetic 3D Cloud Fields: Exploring the Accuracy of two Angular Distribution Models

Nils Madenach[1,2], Florian Tornow[3,4], Howard Barker[5], Rene Preusker[2], and Jürgen Fischer[2]

[1]Leibniz Institute for Tropospheric Research, Leipzig, Germany
[2]Freie Universität Berlin, Berlin, Germany
[3]Columbia University, New York, USA
[4]NASA Goddard Institute for Space Studies, New York, USA
[5]Environment and Climate Change Canada, Toronto, Ontario, Canada

**Correspondence:** Nils Madenach (nils.madenach@tropos.de)

**Abstract.** To accurately estimate outgoing top-of-atmosphere (TOA) shortwave (SW) fluxes from measurements of broadband radiances, angular distribution models (ADMs) are necessary. ADMs rely on radiance-predicting models that are trained on hemispherically-resolved CERES TOA radiance observations. The estimation of SW fluxes is particularly challenging for cloudy skies due to clouds' anisotropy, which substantially varies with their optical properties for any given sun-object-observer geometry. The aim of this study is to investigate, the influence of micro- and macrophysical properties of liquid clouds on SW fluxes estimated by ADMs that are based on a semi-physical model and compare to operational ADMs. We hypothesize that a microphysically-aware ADM performs better in observation angles influenced by single-scattering features.

The semi-physical model relies on an optimized asymmetry parameter $g^\Delta$ that depends on the cloud effective radius. To improve the radiance prediction, $g^\Delta$ is adjusted for the different viewing geometries during the training of the model. In this work these adjustments are linked to single scattering features as the shift of cloud bow and glory with varying cloud droplet size.

For the investigation synthetic 3D cloud scenes based on observations and theoretical assumptions are created. Using a Monte Carlo Model the TOA broad band SW radiances and fluxes of the synthetic cloud scenes are simulated for different scenarios with varying viewing angles ($\theta_v$) along the principle plane and solar angles ($\theta_s$). Analyzing the scenarios the sensitivity and accuracy of the two SW radiance-to-irradiance conversion approaches to cloud droplet size, spatial distribution of liquid water path, and mean optical thickness is quantified.

The study emphasizes that the inclusion of liquid droplet effective radius in the generation of ADMs can result in more accurate SW flux estimates. Particularly for viewing geometries that exhibit single scattering phenomena, such as cloud glory and cloud bow, instantaneous flux estimates can benefit from microphysical-aware ADMs. For instantaneous flux estimates, we found that the error in the SW flux estimates could be reduced by up to 25 $W/m^2$. For cases with very large or small droplets, the median error was reduced by 5 $W/m^2$.



## 1 Introduction

The Earth radiation budget (ERB) quantifies the overall balance of incoming solar radiation and outgoing reflected solar and emitted thermal radiation at the top of the atmosphere (TOA). Quantifying ERB is fundamental for understanding how the climate of Earth will change in the future. The main parameters influencing the ERB are the Earth's surface, clouds, aerosols, and atmospheric gases (e.g. Loeb and Manalo-Smith, 2005; Wild et al., 2014, 2018; Forster et al., 2021). Outgoing radiative fluxes are estimated using, e.g., radiance measurements of broadband (BB) radiometers aboard polar orbiting and geostationary satellites (e.g. Viollier et al., 2009; Dewitte et al., 2008; Velázquez Blázquez et al., 2024a).

However, an accurate estimate of the flux leaving Earth's TOA using only one measurement at a single sun-observer geometry, as is the case for satellites, is challenging. In particular, the reflected solar radiation can be highly anisotropic depending on the observed scene. For clouds, this dependency on the sun-observer geometry is complex and depends on the macro- and microphysical structure of the particles forming the cloud. In past decades, various approaches of radiance-predicting models have been developed and refined to estimate the anisotropy of an observed cloudy scene (e.g., Smith et al. (1986), Loeb et al. (2003), Loeb et al. (2005a), Su et al. (2015), Domenech and Wehr (2011), Tornow et al. (2021), Velázquez Blázquez et al. (2024b)). These so-called angular distribution models (ADMs) can then be used for a radiance-to-irradiance (flux) conversion based on a single observation. An overview of different SW ADM approaches is given in Gristey et al. (2021).

In this study, we investigate TOA SW flux estimates for overcast liquid cloud scenes with varying macro- and microphysical properties. For the SW flux estimates we use the semi-physical log-linear approach (Tornow et al., 2020, 2021), hereafter referred to as the semi-physical approach. Furthermore, we estimate SW fluxes using the sigmoidal approach. This is the currently operational approach for SW flux estimates above clouds (Loeb et al., 2005b; Su et al., 2015), used, e.g., for SW flux estimates from the Clouds and the Earth's Radiant Energy System (CERES). Flux estimates based on the sigmoidal approach are also used as input for the Neural Network used in the EarthCARE processor BMA-FLX (Velázquez Blázquez et al., 2024b). To explore the accuracy of the SW flux estimates and the sensitivity to micro-and macrophysical properties the results of Monte Carlo Simulations (MCS) performed on 125 different 3D-cloud-scenarios are investigated. Other than the sigmoidal approach, the semi-physical approach explicitly incorporates the mean cloud top effective radius ($\overline{r_{eff}}$)) via a parametrized asymmetry parameter ($g^{\Delta}(\overline{r_{eff}})$), Tornow et al. (2020)) that depends on the sun-observer angular bin ($^{\Delta}$). This work explores the following main research questions:

1. How sensitive is the accuracy of TOA SW flux estimates to varying effective radius, cloud homogeneity and optical thickness?

2. Does the explicit incorporation of cloud microphysics in a radiance-to-irradiance conversion approach improve the accuracy of SW flux estimates?

3. Are the adjustments of the asymmetry parameter $g^{\Delta}$ due to the optimization process plausible and what are the specific causes for this?





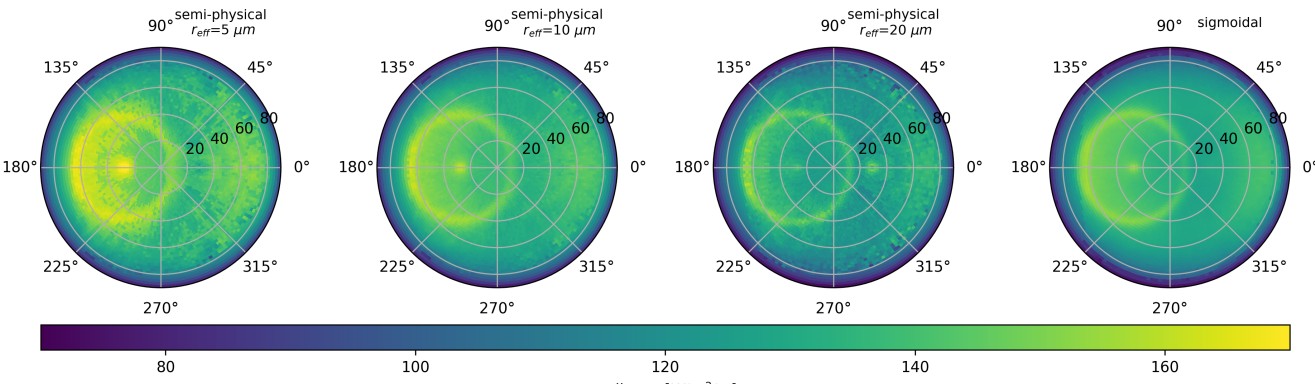

**Figure 1.** Radiance ($\overline{I}$) predicted for a $\theta_s$ of 27 $^\circ$ and an overcast scene over ocean with $\overline{\tau} = 10$. The three panels on the left show simulations using the semi-physical approach with variable $r_{eff}$ and the right panel using the sigmoidal approach.

In Section 2 the theoretical basis of ADMs is described, the creation of the synthetic 3D-cloud scenes is explained and the
configuration of the MCS is given. In Section 3 the results are discussed and in Section 4 the findings are summarized and concluded.

## 2 Theoretical Basis and Methodology

ADMs describe the hemispherically resolved deviation of mean radiance $\overline{I}(\theta_s, \theta_v, \phi)$ reflected by a given scene from the isotropic case. The deviation is expressed through the anisotropic factor ($R$) where values larger than one indicate stronger
reflection than in the isotropic case and vice versa. Equation 1 describes how the anisotropic factor for a given solar zenith $\theta_s$, viewing zenith $\theta_v$ and relative azimuth angle $\phi$ is calculated.

$$R(\theta_s, \theta_v, \phi) = \frac{\pi \overline{I}(\theta_s, \theta_v, \phi)}{\int_0^{2\pi} \int_0^{\frac{\pi}{2}} \overline{I}(\theta_s, \hat{\theta}_v, \hat{\phi}) \cos(\hat{\theta}_v) \sin(\hat{\theta}_v) d\hat{\theta}_v d\hat{\phi}} \tag{1}$$

$$= \frac{\pi \overline{I}(\theta_s, \theta_v, \phi)}{\overline{F}(\theta_s)} \tag{2}$$

For a radiance observation $I_o$ at a given sun-observer geometry the TOA flux estimates ($F(\theta_s)$) is calculated using the
anisotropic factor derived in Equation 1 following Equation 3.

$$F(\theta_s) = \frac{\pi I_o(\theta_s, \theta_v, \phi)}{R(\theta_s, \theta_v, \phi)} \tag{3}$$

**Semi-physical approach**

The semi-physical approach uses explicitly the microphysics of clouds and the water vapor load above the clouds (above cloud water vapor, ACWV) to predict the hemispherical field of outgoing TOA SW radiances $\overline{I}$ (see Fig. 1. The TOA SW anisotropy





is sensitive to both variables and in case of effective radius this can lead to anisotropy differences of up to 8 % (Tornow

et al., 2018). To incorporate these dependencies, the semi-physical approach uses a simple model (Eq. 4) relating the outgoing

radiance $I$, to the incoming solar irradiance ($S_0$) via a footprint albedo $\alpha$ and a factor describing the attenuation due to water

vapor above clouds ($e^{-2ACWV}$). The factor 2 arises from the fact that the light passes the water vapor layer twice before

reaching the TOA. By using the logarithm (Eq. 5) the model becomes linear and a simple first-degree polynomial function

(Eq. 8) can be fitted to the observations.

$$I(\theta_s, \theta_v, \phi) \sim S_0 \cdot \alpha \cdot e^{-2 \cdot ACWV} \tag{4}$$

$$\log I(\theta_s, \theta_v, \phi) \sim \log S_0 + \log \alpha - 2 \cdot ACWV \tag{5}$$

The footprint albedo $\alpha$ (Eq. 6) is the sum of the clear sky portion of the scene ($f_0$) multiplied with the clear sky albedo ($\alpha^{cs}$)

and the cloud fraction ($f_1$) multiplied with the two-stream albedo ($\alpha_{ts}$). $\alpha_{ts}$ (Eq. 7) depends on the footprint mean cloud optical

thickness ($\overline{\tau}$) and the parametrized asymmetry parameter for the given sun-observer bin ($g^\Delta(\overline{r_{eff}})$).

$$\alpha = f_0 \cdot \alpha_{cs} + f_1 \cdot \alpha_{ts} \tag{6}$$

$$\alpha_{ts} = \frac{\left(1 - g^\Delta(\overline{r_{eff}})\right) \cdot \overline{\tau}/2}{1 + (1 - g^\Delta(\overline{r_{eff}})) \cdot \overline{\tau}/2} \tag{7}$$

The asymmetry parameter is a function of the cloud microphysics represented via the footprint mean cloud top effective radius

($\overline{r_{eff}}$). During the training of the model, the asymmetry parameter has been optimized for each sun-observer geometry bin

($\Delta$) to improve the linearity between the observed radiances (see e.g., Fig. 7 lower panel). The bin wise optimization of the

asymmetry parameter $g^\Delta(\overline{r_{eff}})$ the $\alpha_{ts}$ accounts for various 3D-effects that the model does not account for as well as single

scattering features caused by the underlying phase function as, e.g. the widening and shift towards the forward direction of

cloud glory and a shift towards the direct backscatter of the cloud bow with smaller $\overline{r_{eff}}$ (e.g., Mayer et al. (2004)). After

the optimization the single scattering features became apparent in the modeled radiances (see Fig. 6). An in depth explanation

and further discussion of the semi-physical approach are given in Tornow et al. (2018) and Tornow et al. (2020). For our

investigation we will focus on overcast scenes ($f_1 = 1$) above ocean with $f_0 = 0$ and $ACWV = 0$. In order to provide angular

coverage for the ADM construction, observations between 2000 and 2005 were used when CERES measured in the rotating

azimuth plane scan mode. The CERES Ed4SSF (Edition 4.0 Single Scanner Footprint) dataset of Aqua and Terra ( described

in Su et al. (2015))) that combines MODIS and CERES L2 data has been used to collect observations for sun-observer bins

($\Delta$) of 2°×2°. To fit the collected data to the semi-physical model, an ordinary-least-square method has been used with the free

parameters A, B, C (Eq. 8). Further explanation can be found in Tornow et al. (2020).

$$\log I(\theta_s^\Delta, \theta_v^\Delta, \phi^\Delta) \approx A + B \cdot \log \alpha + C \cdot ACWV \tag{8}$$

**Sigmoidal approach**

The sigmoidal approach (Loeb et al., 2005b; Su et al., 2015) uses information of the CERES footprint average cloud optical

depth $\overline{\tau}$ (exponential of the average over logarithmic $\tau$ values) and the cloud fraction $f$. Per angular bin a sigmoidal function





**Table 1.** Values used for the creation of cloud vertical profiles.

| $\gamma$ | $\Gamma$ | $k$ | $A$ |
|---|---|---|---|
| 1 | 1.5 | 0.8 | 0.0145 |

(9) is fitted to the observed CERES radiance $I$ and $x = log(f \cdot \overline{\tau})$. Where $I_0$, $a$, $b$, $c$, and $x_0$ are free parameters.

$$I = I_0 + \frac{a}{[1 + e^{-(x-x_0)/b}]^c} \tag{9}$$

To explore the research questions raised above, 125 realistic 30x30 $km^2$ 3D-cloud scenes with a horizontal resolution of 1 $km$ and varying mean optical thicknesses ($\overline{\tau}$), homogeneities ($\nu$) and droplet number concentrations ($N_d$) are generated. The exact 
procedure is explained below.

## 2.1  Brief Recap on Cloud Adiabatic Theory

Following the adiabatic theory described, for example in Brenguier et al. (2000) and Wood (2006), the vertical profile of cloud liquid water content (LWC) can be approximated by Equation 10. The mean cloud volume radius ($r_{vol}$) for a given layer depends on the amount of liquid water in the layer LWC and the concentration of cloud droplets ($N_d$). The effective radius 
($r_{eff}$) is related to $r_{vol}$ via the constant $k$ (see Equation 12).

$$\text{LWC}(z) = \gamma \cdot \Gamma \cdot z \tag{10}$$

$$r_{vol} = \frac{3\text{LWC}}{4\pi \rho_w N_d} \tag{11}$$

$$r_{eff} = k^{-1/3} \cdot r_{vol} \tag{12}$$

Where $z\,[m]$ is the height from cloud base, $\gamma\,[1]$ the degree of adiabaticity, $\Gamma\,[g/m^3/m]$ the adiabatic rate of increase of liquid 
water content and $N_d\,[1/cm^3]$ the cloud droplet number concentration. Following the equations above, the optical thickness $\tau$ of the cloud depends only on the geometrical thickness $h$ and $N_d$ (see Equation 13).

$$\tau(h) = A \cdot (\gamma \cdot \Gamma)^{2/3} \cdot (k \cdot N_d)^{1/3} \cdot h^{5/3} \tag{13}$$

where $A$ is a constant. The values used for the parameters and constants are shown in Table 1.

## 2.2  Creation of Synthetic 3D-Cloud Scenes

For the creation of the scenes we analyzed a MODIS frame from September 5th 2014 above the south-east Atlantic covered with marine boundary layer Stratocumulus clouds. To obtain realistic ranges of mean cloud optical thickness $\overline{\tau}$ and homogeneities $\nu$ in marine boundary layer clouds, we separated the MODIS frame into 30x30 $km^2$ boxes as shown in Fig. 2 (a). For each



box, we calculated $\overline{\tau}$ following Equation 14 and $\nu$ following Equation 15.

$$\overline{\tau} = \exp(\overline{\log(\tau)}) \tag{14}$$

$$\nu = \left(\frac{\overline{\tau}}{std(\tau)}\right)^2 \tag{15}$$

For the boxes within the MODIS frame, we found values $\overline{\tau}$ values between 2.8 and 20.1 and $\nu$ values between 2 and 26.

For the creation of idealistic cloud scenes, gamma functions based on Barker et al. (1996) have been used to calculate PDFs of optical thickness values for the given $\overline{\tau}$ and $\nu$ pair of the scene (see Fig. 2 (b)). In total 25 PDFs have been created based on five $\overline{\tau}$ values (2.8, 4.5, 7.4, 12.2, 20.1) and five $\nu$ values between 2 and 26. In the next step, the idealistic range of $\tau$ values

(p<0.001) for the given $\overline{\tau}$ and $\nu$ was extracted from each PDF and the cloud geometrical thickness $h$ using Equation 13 was calculated for each $\tau$ bin within the range.

Assuming a constant cloud base and $N_d$, the optical thickness $\tau$ depends only on the cloud top height (Eq. 13). Using Equation 10 to 12 vertical profiles of LWC and $r_{eff}$ are calculated for grid-cell of the scenarios. The vertical resolution of the profiles is set to 25 $m$. For the stratocumulus deck of the MODIS frame (Fig. 2 a)), a very similar cloud base is assumed. Using

radiosonde measurements of $T$, and water vapor at 12 p.m. LT in St. Helena, a cloud base of 761 $m$ was assumed (see Fig. 2 (c)). To obtain a realistic spatial distribution of the cloud profiles, we assigned them to MODIS boxes with similar $\overline{\tau}$ and $\nu$ and used the spatial distribution of the $\tau$ values within the box to assign the profiles.

Applying five different $N_d's$ of 25, 50, 100, 200 and 400 $1/cm^3$, resulted in 125 scenes. Figure 3 and 4 display the results, showing the geometrical thickness of the clouds for the range of $\overline{\tau}$ and $\nu$ and for a $N_d$ of 25 $[1/cm^3]$ and 400 $[1/cm^3]$ re-

spectively. Comparing Fig. 3 and Fig. 4 we see, that to reach the same $\tau$ the cloud extend must be larger for larger cloud droplets (smaller $N_d$). This is because smaller particles (larger $N_d$) scatter the light stronger in the backward direction, leading to stronger attenuation.

## 2.3 Monte Carlos Simulations

The 125 synthetic scenes of cloud fields are used as inputs for a Monte Carlo Model (Marchuk et al., 1980; Barker et al.,

2003) to simulate the SW TOA radiances and calculate TOA SW fluxes. For the simulations cyclical boundary conditions are assumed. For each scene, simulations for 40 viewing zenith angles between -77 to 77° along the principle plane and for solar zenith angles of 1, 27, 55, and 75° has been performed. In total, this resulted in 20 000 scenarios. For each simulation, $10^7$ photons have been used. The ocean surface is Lambertian (corresponding to a near-surface wind speed of 0 $m/s$ in ADMs) with wavelength independent albedo of 0.05. For the hydrometeors Mie-phase functions with 1800 angular bins are used.

In order to investigate the contribution of single scattering to the outgoing TOA radiance (research question 3), a histogram of the weighted fraction of the number of scattering events has been stored from the simulation for each scenario.





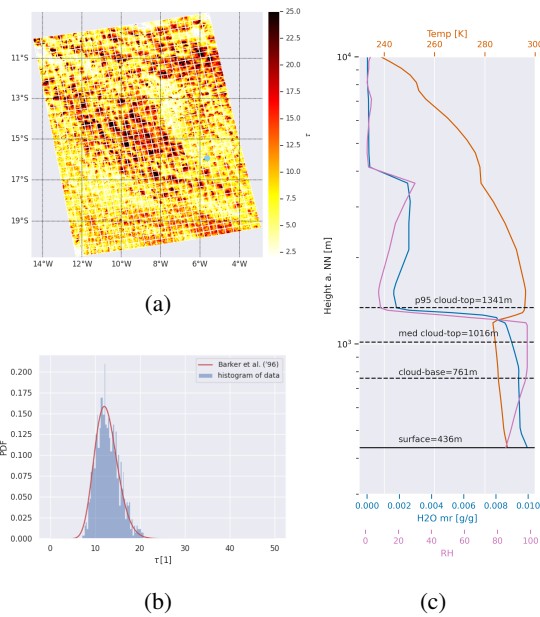

**Figure 2.** (a) MODIS retrieved $\tau$ above south-east Atlantic on September 5th, 2014. The frame is separated into 30x30 $km^2$ boxes. The blue dot denotes the location of St. Helena. (b) Example of a gamma distribution of $\tau$ (red line) following Barker et al. (1996) generated using $\overline{\tau} = 12.6$ and $\nu = 24.1$ from a 30x30 $km^2$ box of the MODIS frame. In blue the histogram of the $\tau$ values within the box is displayed. (c) Vertical profiles of $T$, relative humidity ($RH$) and $H_2O$ mixing ratio from radiosonde ascent at St. Helena from September 5th, 2014 at 12 p.m. LT. The dashed black lines indicate the cloud base and the 50th and 95th percentile of cloud tops for all 125 scenes.

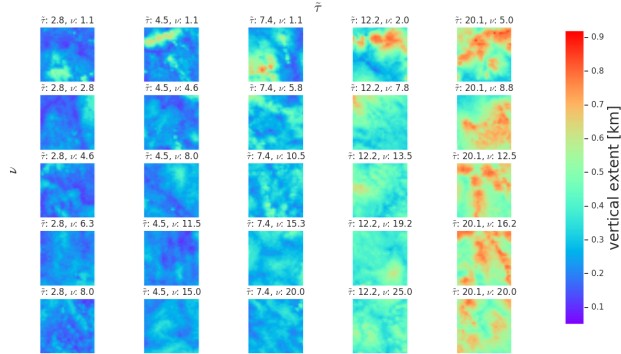

**Figure 3.** Cloud geometrical thickness for the 25 scenes with varying $\overline{\tau}$ and $\nu$ and with the $N_d$ set to 25 $[1/cm^3]$

## 3 Results

In addition to flux estimates based on ADMs using the semi-physical approach, we use flux estimates based on ADMs using the currently operational sigmoidal approach, reconstructed in Tornow et al. (2021). In contrast to the semi-physical approach, the



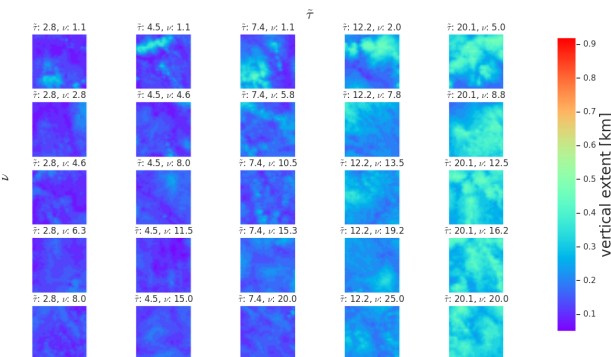

**Figure 4.** Same as Fig. 3 but with $N_d$ of 400 $[1/cm^3]$

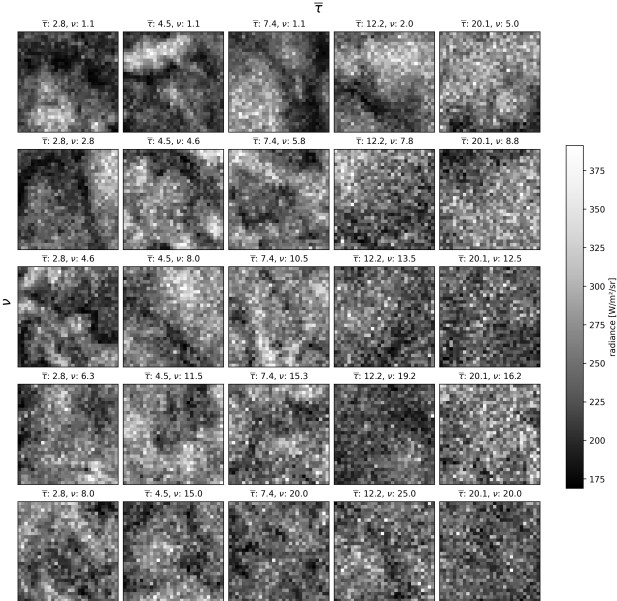

**Figure 5.** Monte Carlo simulations of TOA radiances for scenes with $N_d = 400$ and for $\theta_v = -29°$ and $\theta_s = 27°$.

sigmoidal approach does not explicitly take into account the cloud droplet effective radius. Further description of this method can be found, for example, in Loeb et al. (2005b), Su et al. (2015) and Gristey et al. (2021). In Fig. 5 an example of the results from the Monte Carlo Simulations is shown. Using the scene averaged radiances as $I_o$ (see Fig. 6), fluxes are estimated for all scenarios using both approaches.

Figure 6 displays the mean radiance for one scene ($\overline{\tau} = 2.8$ and $\nu = 8$) at $\theta_s = 1°$ and for different $N_d$. We see large differences in the reflected radiation by only changing the microphysical property of the mean cloud droplet size. We also observe single scattering features become apparent, such as the broadening of the cloud glory and the shift of the cloud bow towards the direct





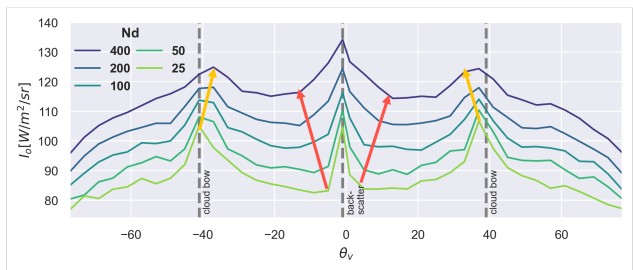

**Figure 6.** Mean radiance of the MCS used as $I_o$ along $\theta_v$ (principle plane) and for varying $N_d$. For a $\theta_s = 1°$ and for the scene with $\overline{\tau} = 2.8$ and $\nu = 8$. The vertical lines indicate the location of single scattering phenomena as the cloud bow and cloud glory (around the direct backscatter). Yellow arrows indicate the shift of the cloud bow towards the direct backscatter and red arrows the widening of the cloud glory with smaller droplets.

backscatterer with decreasing droplet size (increasing $N_d$).

The upper panel of Fig. 7 shows the TOA SW fluxes across the principle plane for scenarios with a solar zenith angle of $1°$ and scenes with a optical thickness of $\overline{\tau} = 7.4$ and homogeneity of $\nu = 20$. The solid lines show the "true" fluxes from the

simulations. The dashed lines represent the flux estimates based on the semi-physical approach and the dotted lines the flux estimates based on the sigmoidal approach. The colors of the lines illustrate estimates for scenes with different droplet number concentrations $N_d$. In addition, the colored dots indicate the weighted fraction of photons of the scenario that experienced single scattering before reaching TOA. The lower panel illustrates the parameterized asymmetry parameter $g^\Delta$ used for the semi-physical ADMs. The vertical lines show the location of the cloud bow and the direct backscatter, around which the cloud

glory forms. The results show that an increase of $N_d$ from 25 to 400 $1/cm^3$ produces up to 100 $W/m^2$ higher fluxes at TOA. Around the direct backscatter direction, where the cloud glory contributes to the observed radiance of the scenes, the ADMs created using the semi-physical approach result in flux estimates closer to the MCS. This is also the case at the cloud bow around +-40°. Especially for high droplet number concentrations (small droplet sizes), where the enhanced reflectance due to single scattering effects are largest, the currently operational approach underestimates the fluxes. Due to the bin-wise parametrized

asymmetry parameter (bottom panel), the semi-physical approach is able to capture single scattering features as the widening and shift towards the forward direction of the cloud glory as well as the shift towards the direct backscatter of the cloud bow with increasing $N_d$ (Tornow et al., 2021). The results illustrate that this leads to more accurate flux estimates in these geometries in comparison to the sigmoidal approach.

For angles influenced by the cloud bow or cloud glory, an enhanced contribution of single scattering (colored dots) to the

reflected radiance is clearly visible. Furthermore, the single scattering fraction increases with increasing $N_d$. This underpins the hypothesis, that the adjustments of $g^\Delta$ (lower panel) during the bin wise optimization procedure (Tornow et al., 2020), are primarily due to perceived single scattering effects as, e.g., the widening and shift of the cloud glory and the shift of cloud bow. Figure 8 shows the results for $\theta_s$ for $27°$ with similar results. At geometries influenced by the cloud glory, the sigmoidal approach performs well under average microphysical conditions (e.g, $N_d = 100$) but overestimates at low and underestimates at



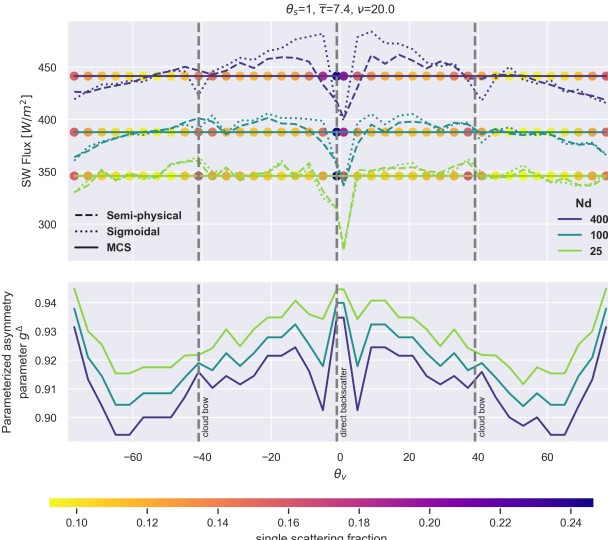

**Figure 7.** Upper panel: Flux estimates using ADM based on the semi-physical (dashed) and based on the sigmoidal (dotted) approach for different droplet number concentrations along the principle plain. The true flux of the Monte Carlo Simulations (MCS) is shown in the solid line. The colored dots represent for each scenario the fraction of photons that has been scattered only once. Lower panel: parametrized asymmetry parameter $g^{\Delta}$ used for semi-physical approach.

high $N_d$ the fluxes. This agrees with Fig. 1 where the predicted radiances using the sigmoidal approach (right panel) correspond well with the semi-physical approach using a typical mean effective radius of 10 $\mu m$ (second panel) but is lower for small droplets (first panel) and higher for large droplets (third panel).

At high and low $\theta_v$ both approaches struggle because fewer training data were available for these geometries. In sun-glint affected angles both approaches have problems in accurately estimate the fluxes, but the semi-physical approach generally deviates even more. These findings are consistent with larger uncertainties of the models for these geometries found in Tornow et al. (2020). In Fig. 9 the flux deviations for scenes with a $N_d$ of 400 $1/cm^3$ and the corresponding highest homogeneity are illustrated for varying $\overline{\tau}$ and $\theta_s$. The upper panel shows the deviation of flux estimates using semi-physical and the lower panel using sigmoidal based ADMs. In the top of each panel the mean over all $\overline{\tau}$ (black line) and the standard deviation (gray shadows) are shown. The doted isoline indicates areas where the differences exceed the EarthCARE mission goal of $\pm 10$ $W/m^2$.

We see that generally the semi-physical approach deviates less, especially for viewing-geometries around the cloud glory and cloud bow. This is the case for all optical thicknesses and solar zenith angles, but for small optical thicknesses the effect is more distinctive. The deviations in instantaneous flux estimates using the semi-physical approach compared to the currently operational approach can be reduced by up to 25 or even more $W/m^2$. In the sun glint affected regions both approaches show large uncertainties. Both absolute and relative deviations (not shown) increase with smaller $\overline{\tau}$ in this region. This indicates that even in overcast cases with high optical thicknesses, the sun-glint significantly influences the TOA radiances and the




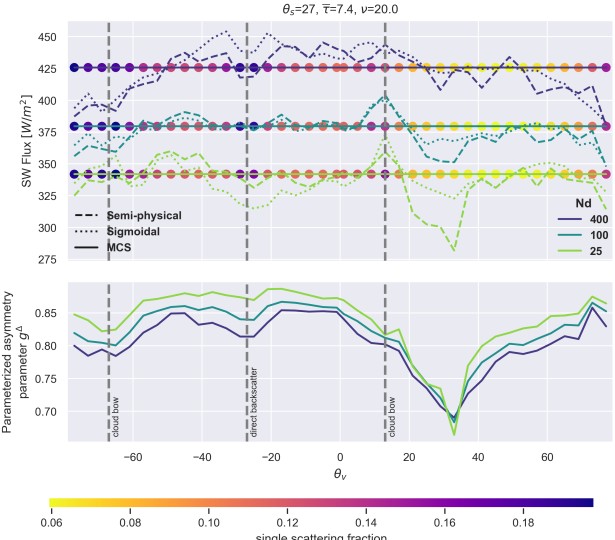

**Figure 8.** As Fig. 7 but for a $\theta_s$ of $27^\circ$

ADMs are not able to accurately account for this. Figure 10 illustrates the box-whisker plots ($n$=2000) of flux deviations ($\Delta F$)

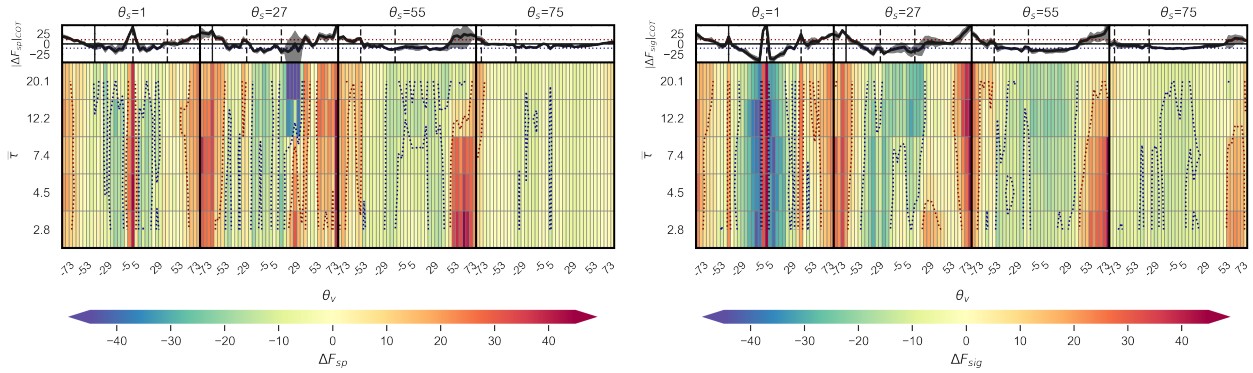

**Figure 9.** Deviation of fluxes estimated ($\Delta F = F_{MCS} - F_{est}$) using ADM's based on the semi-physical approach (upper panel) and on the sigmoidal approach (lower panel) from the fluxes of MCS along the principle plane and for varying $\theta_s$ and $\overline{\tau}$. The $N_d$ of the scenes is 400 [$1/cm^3$] and for $\nu$ always the scene with the highest homogeneity is selected. The dotted blue and red lines represent the -10 and 10 $W/m^2$ threshold. On top of each panel the deviation averaged over all optical thicknesses is shown. The shaded areas mark the 5th and 95th percentiles. The vertical lines indicate the direct backscatter and the cloud bow ($\theta_s \pm 40^\circ$).

for scenarios in the backward (left) and forward direction (right) and for different droplet number concentrations. The flux estimates based on the semi-physical approach deviates from the simulations in the backward direction. Especially, at extreme
$N_d$ (e.g., 5 $cm^{-3}$ and 400 $cm^{-3}$) the semi-physical approach deviates less from the simulations in the backward direction. The





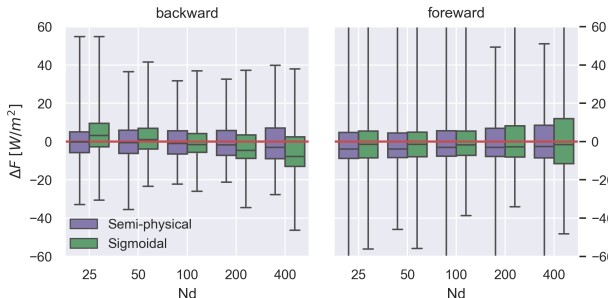

**Figure 10.** Box-Whisker plots showing flux deviations using $n = 2000$ scenes of all $\theta_s$, $\overline{\tau}$, $\nu$ and for the left panel of $-77 < \theta_v < 0$ and for the right panel $0 < \theta_v < 77$.

median deviation of the 2000 scenarios can be reduced by up to 5 $W/m^2$. This can be explained by the fact that the sigmoidal approach does not explicitly take $N_d$ into account. By using observations independently on their microphysics, the sigmoidal approach produces best estimates at average $N_d$ of, e.g. 50 and 100 $1/cm^3$, but is less accurate in extreme high or low $N_d$s. Table 2 lists the probability of $\Delta F > 10\,W/m^2$ for scenes with different $N_d$ and for scenarios calculated for the Backward, Forward and Nadir direction. For high $N_d$ the semi-physical approach can reduce the probability by up to 20 %. We found that

| Nd | Forward | Backward | Nadir |
|---|---|---|---|
| 25 | 35.2/34.1 | 23.0/31.5 | 39.5/38.5 |
| 50 | 33.7/31.0 | 27.0/23.9 | 41.0/33.0 |
| 100 | 34.0/31.5 | 27.1/24.3 | 37.0/31.0 |
| 200 | 36.7/42.8 | 31.0/35.1 | 38.0/43.5 |
| 400 | 43.8/58.2 | 41.0/55.8 | 58.5/79.5 |

**Table 2.** Table showing the probability in % of $\overline{\Delta F > 10\,W/m^2}$. Forward includes all scenarios of $0 < \theta_v < 77$, backward of $0 > \theta_v > -77$ and nadir of $-1 < \theta_v < 1$. The purple numbers are for the semi-physical approach and green numbers for the sigmoidal approach.


in general the mean absolute relative error decreases with increasing $\overline{\tau}$ and $\nu$ and increases with increasing $\theta_s$ and especially for the sigmoidal approach with increasing $N_d$ (not shown).

In Fig. 11 the $30 \times 30 km^2$ scene has been divided into subdomains with sizes of 25, 20, 15 and 10 $km$. The variability in $\Delta F$ increases with smaller domain sizes. As in the case of 30 $km$, the semi-physical approach produces better estimates in 215 the backward direction and for domains with extremely high or low $N_d$ for all resolutions. For a domain size of 10 $km$ both approaches show a positive bias. EarthCARE's assessment domain has also a size of 100 $km^2$. The results might be of interest for the validation of the BMA-FLX product.





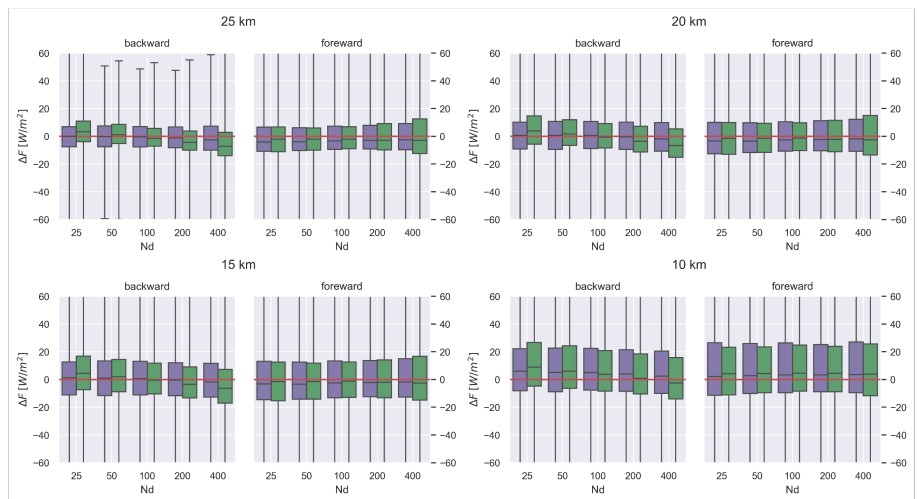

**Figure 11.** As Fig. 10 but for domain sizes of 25, 20, 15 and 10 $km$.

## 4 Conclusion and Discussion

In this study, Top of Atmosphere (TOA) short wave (SW) radiances are simulated for 125 synthetic 3D cloud scenes with
varying cloud optical thickness, cloud homogeneities and cloud droplet number concentration. Seeking TOA SW flux closure
above liquid clouds, two ADMs namely the semi-physical (Tornow et al., 2021) and sigmoidal approach (Su et al., 2015), are
compared against the fluxes calculated using the Monte Carlo Model.

We found that in general the mean absolute relative error decreases with increasing $\overline{\tau}$ and $\nu$ and increases with increasing $\theta_s$
and especially for the sigmoidal approach with increasing $N_d$ (Research question 1).

The microphysical-aware semi-physical approach reduces the errors in instantaneous flux estimates by up to 25 $W/m^2$
compared to the sigmoidal approach. The improvements are found to be largest for geometries in the backward direction and
for scenes where microphysics deviates most from mean conditions as e.g., for extremely high or low $N_d$. The median deviation
($n = 2000$) of these scenarios (backward direction and different $N_d$) is improved for all $N_d$ by up to 5 $W/m^2$. The results are
in agreement with a study from Tornow et al. (2021) comparing the two radiance-to-irradiance approaches using satellite data
(Research question 2).

Analyzing the Monte Carlo Simulations (MCS), the adjustments of $g^\Delta$ used in due to the optimization could be related to
changes in the fraction of single scattering events contributing to the TOA radiance signal. The changes in the single scattering
fraction are associated with phenomena such as cloud bow or cloud glory that depend on cloud microphysical properties
(Research question 3).

By explicitly incorporating cloud microphysical properties, through the effective radius, the semi-physical approach substan-
tially reduces errors in flux estimates, particularly in scenarios affected by single scattering phenomena, such as cloud glory
and cloud bow. As these phenomena are strongest in the backward direction, the improvements compared to the currently



operational approach are most pronounced in scenarios with corresponding angles. For sun-glint affected observations, the flux estimates show large variabilities and larger errors and should therefore be interpreted with caution.

Using the optimized $g^\Delta$, the semi-physical approach is able to capture the shift of the cloud bow and the widening of the cloud glory with decreasing droplet size, explaining the more accurate estimates in these geometries.

The findings in this study encourage the additional use of the semi-physical approach for the EarthCARE mission launched in May 2024 (Wehr et al., 2023). The mission performs a radiative closure experiment by comparing simulated fluxes, based on measurements from active and passive instruments aboard the satellite, with estimated fluxes, based on measurements from

a broadband radiometer (Velázquez Blázquez et al., 2024b). Especially for viewing geometries in the backward scattering direction, the semi-physical approach might help to reduce misinterpretations of flux deviations in the closure that are larger than the mission goal of $10\ W/m^2$, but that actually are due to uncertainties in the SW flux estimates. When seeking the radiative closure for EarthCARE, we should always be aware that the BBR-based fluxes are estimates and not measurements and have uncertainties that vary depending on surface type, atmospheric conditions, and sun-satellite angles. Especially for

high $N_d$ the semi-physical approach reduces the probability of $\Delta F > 10\ W/m^2$ substantially. For both approaches, the flux uncertainties increased with smaller footprint size.

Overall, the study indicates that incorporating information on cloud microphysical properties into the development of ADMs is a promising pathway for enhancing TOA SW flux estimates above clouds. In contrast to previous ERB missions, which primarily focused on minimizing the global bias of flux estimates, EarthCARE is designed with an emphasis on radiative

closure and may particularly benefit from microphysical-aware instantaneous flux estimates. The results presented encourages further study on e.g., sensitivity to the accuracy of the physical retrievals, to noise, and observational conditions.

*Author contributions.* FT, NM and HB designed the study. NM created the input scenes, performed the analysis and wrote the paper. HB carried out the Monte Carlo Simulations. RP and JF supervised the work.

*Competing interests.* The authors declare that they have no conflict of interest.

*Acknowledgements.* This research has been supported by the European Space Agency (ESA; contract nos. 4000112019/14/NL/CT (CLARA), 4000134661/21/NL/AD (CARDINAL), and 4000133155/20/NL/FF/tfd (ICERAD)).



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





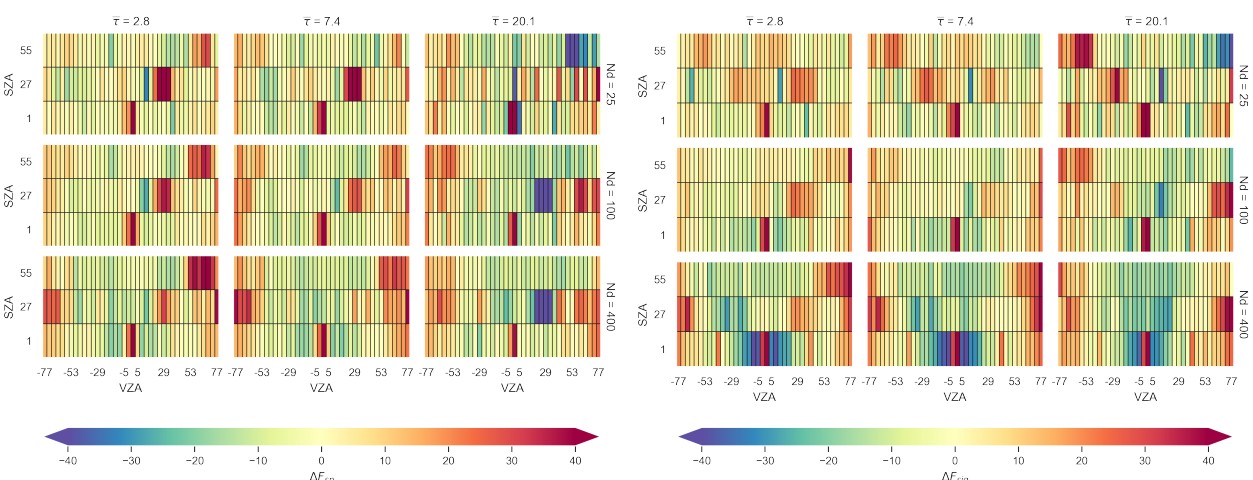

**Figure A1.** Flux deviations for scenarios with varying $\theta_s$, $\theta_v$, $N_d$ and $\overline{\tau}$. The upper panel shows the results for the semi-physical approach and the lower panel the results for the sigmoidal approach.