# Peer review of "Seeking TOA SW Flux Closure over Synthetic 3D Cloud Fields: Exploring the Accuracy of two Angular Distribution Models"

_EGUsphere, 2025_

## Referee Comment (RC2)

**Summary**

The paper considers the improvement in accuracy of instantaneous flux estimates for cloudy conditions using empirical ADMs based on two different methods of defining and assigning the ADMs.  Instantaneous flux accuracy is of increasing interest and of particular relevance to the EarthCARE BBR instrument for example, as well as to any application that might make use of instantaneous flux retrievals. However, traditional SW ADMs used operationally often prioritize minimising global bias over maximising instantaneous accuracy and thus are not optimized for maximising the latter. Therefore, the work addresses an important problem of interest to the community interested in using instantaneous fluxes particularly for cloud study.

To investigate the problem, 3D radiative transfer simulations using the Monte Carlo method are used informed by cloud inhomogeneity derived from MODIS observations. This seems a reasonable approach in principle and a good attempt to deal with the complexity of 3D effects and cloud in-homogeneity in a realistic manner. However, it should be made clearer in the work that this is a limited case-study that is not sufficient to make broad globally applicable claims about the relative merits of the two methods but rather represents a first step in exploring their merits for a particular case.

I think that the intent of the paper is valuable, and the basic tools used are appropriate, the work is well laid out and generally well written. However, I think there are some major aspects regarding the application which are fundamental to the validity and usefulness of the comparison and the interpretation of the result that need to be addressed before the work is suitable for publication. Some of this relates merely to further clarification and more consideration of how the results are presented and summarized. However, I also have some concerns over the realism of the simulations used and how the retrieved fluxes are derived from them which I think needs further support or modification.

**Major concerns**

**The application of the two methods.**

**Section 2 lines 58 to 105**: Clarification needs to be included in the text as it is not clear from the current description if the method which uses what is called the 'sigmoidal approach' uses the CERES operational ADM's themselves (as described in lines 99 to 102), uses ADMs based on the same CERES observational period described for the semi-empirical approach (lines 92 to 97) or is in this case based on the simulations used to 'explore the research questions' as discussed in lines103-105. Furthermore, it is not clear if the full methodology for the 'sigmoidal approach' described in Loeb 2005 (a & b) is employed here. Specifically sigmoidal fits only used for thick clouds.  My assumption is that the operational CERES ADMs are the basis for comparison including all their variations but please clarify, I think the problem is the that the method refers to the approach rather than a specific set of ADMs.

**Section 3**. It is unclear when the two methods are applied to the simulated data how the required parameters to choose the ADMs are determined. For example, for the 'sigmoidal approach' what optical depth is used? Is it a value derived from a MODIS-like retrieval applied at the relevant resolution and wavelength on the simulated radiances, or is it taken from the input parameters to the simulation – in which case at what wavelength and is this a fair test? If the radiances simulated differ from those used to derive the observational ADMs, it would stand to reason that the associated narrow band radiances which form the basis of the MODIS derived optical depth by which they are classified would also differ, and thus the optical depth retrieved from MODIS may not be the same as that used in the simulation. It is however the MODIS retrieved optical depth that needs to be used to determine the appropriate anisotropy.

Similarly for the semi-empirical approach where does the cloud microphysical and above cloud water vapour information come from (I know the latter is set to zero in the simulations is this taken as a given in the ADM choice?) and is this consistent with how this method will be applied in practice. Again, if the effective radius is to be retrieved from MODIS it should be retrieved by that method from the simulated radiances.

Finally, it is of significant relevance to the accuracy of application in an operational context how accurately the parameters required to select the ADM can be retrieved, and how sensitive the resulting flux is to errors in their retrieval. These aspects are not considered here or indeed even mentioned. As more parameters are required to apply the semi-physical approach this is potentially a bigger problem for this case and should be at least mentioned in any comparison of the two methods.

**The simulations used for the test**

**Section 2.1 to 2.3.** The CERES ADMs are empirically derived from observations and represent real world conditions. The distribution of cloud properties within any bin is expected to be representative of the real-world distribution of these properties and the effects of the surface reflectance and background aerosol will also be implicitly included. These points also apply to some extent to the semi-empirical approach as this is also based on observations although the issue may be lessened by the finer division of scenes reducing the dependence of the result on the distribution found in nature.

Given this, the realism of the test and understanding how this relates to the real world range and frequency of cloud properties is fundamental to providing a useful evaluation. As a minimum, we need to know that the cloud properties used are realistic and how common they are. It seems quite strange to me given the empirical nature of the ADMs tested and the significant amount of observational data used to derive them, that although MODIS data is used to look at optical depth and homogeneity it is not used to for the effective radius which rather uses a fixed relation to the optical depth and varied to various constant values between scenarios only (via variation in Nd). It is not clear to me that the relationships used in section 2.1 are at all valid for the case used, they seem to employ several assumptions that will not be

universally true over a range of reff or wavelength (extinction efficiency of 2 for example which is a large particle approximation). Furthermore, the origin of the Ndues val chosen are not explained. I am also confused by the Wood 2006 reference here which has no journal, publisher or doi associated with it, is this a technical note a chapter from a book please can you clarify this reference, the equations stated seem to come from this form, I don't think the Qext = 2 simplification is a feature of the other reference. Apologies if I have misunderstood but forcing this fixed relation to reff seems to limit the realism of the simulation that was so carefully ensured for the optical depth variation. For the MODIS scene evaluated optical depth variations are attributed solely to geometric thickness variations and the value of Nd with each case have a single reff. Does the MODIS data show reff inhomogeneity as well as optical depth inhomogeneity or does it corroborate the constant values assumed here? In the discussion I think it would be helpful to translate to reff rather than Nd as reff is the parameter both retrieved by MODIS and required to apply the semi-physical approach.

The realism of other aspects of the simulation including the surface and the intervening atmosphere is also relevant. For example, if the simulations set the above cloud water vapour to zero and this can be selected as a case for the semi-physical retrieval this could be unfair for the sigmoidal retrieval. If zero above cloud water vapour is unrealistic or an outlier the simulations don't represent the real-world values implicitly included in the sigmoidal ADMs, thus we need to know how big an effect this discrepancy is. For the surface a Lambertian ocean surface seems quite limited, I am confused as to why this is stated as being equivalent to a wind speed of zero as I would have thought reflection from calm ocean is more likely to be considered as a specular rather than Lambertian reflector. How important this assumption is in taking the simulations out of the realm of realism and therefore presenting an unfair test of observational ADMs needs to be considered.

**The details of the method and the application of the test case**

A little more detail on the 'sigmoid approach' would be sensible to include here. Specifically what data are used as the basis here, what are the fitted parameters and what special treatments (for example for thinner cloud) applied. Similarly for the semi physical approach a bit more detail on the parameters used and how their values are obtained both operationally and in this case would be helpful. The use of a view-angle dependent asymmetry parameter probably requires some specific explanation here as it is a rather unusual choice specific to a particular implementation of the semi-physical approach and is contrary to the normally understood meaning of an asymmetry parameter (which pertaining to a two-stream approximation would have no view angle dependence).

**The highlighted comparisons (section 3)**

**Figure 6 and associated discussion.** It is not clear to me that the radiance comparison plots shown in figure 6 are the best choice when the comparison is concerned with the improvement in the derived flux from the radiance. Whilst obviously related would not a comparison of the anisotropy be a way to display more relevant and complete information here?

**Figure 6 and 7** I would consider the use of 1 degree SZA a very limited case, which might be sensible to include in the simulations to cover the range but would not seem the best choice for plotting examples as in figure 6 and figure 7 for general discussion, a solar zenith angle of 55 would possibly be a better more general choice.

**Figure 6, 7, 8 and 9.** Similarly, it is not clear that concentrating on the principal plane is particularly helpful unless you expect the flux retrievals to be more commonly associated with the principal plane. Plots showing the full space of the anisotropy or flux difference as used to show the radiance distribution in figure 1 maybe more helpful. Alternatively, following the current format but adding an indication of the range for the points outside the principal plane would be an alternative.

**Figures 9, 10 and 11 and associated discussion in sections 3 and 4.** Summary statistics for 2000 scenarios are shown, and the median used as a primary comparator for the performance of the two methods. I assume that the 2000 scenarios arise from the division of the original 20,000 scenarios into forward and backward directions and the 5 Nd and thus comprise 25 optical depth PDFs, 20 viewing angles and 4 solar zenith angles each. It is not clear if any weighting is applied to these 2000 cases to make them a reasonable representation of the frequency of the scenarios to be encountered. For example, does the result equally weight the solar and viewing angles in deriving the summary statistics or are they weighted according to their likely frequency of occurrence in some dataset and if so what dataset or is some angular integration done to derive the final result. Similarly, is any weighting given to the different optical depth PDFs and if so, is this based on the single case study analysed or given more global consideration.

**Figure 11 and associated discussion**. Dependence on domain size is briefly addressed, the inherent variation of domain size with viewing angle in the observations is not considered and should at least be mentioned here. It is inherent in the empirical ADMs.

**Conclusions (section 4)**

It might be helpful to restate the research questions in the conclusion. Research question 1 pertains to reff and is answered as Nd. Given the relation between these is buried in detailed assumptions in the main body of the work this should be translated here (and in the analysis) to properly answer the research question posed.

In reference to all the research questions and final recommendation the conclusions need to acknowledge, even assuming that the simulations are realistic, they represent a single case study. This is sufficient to highlight the need for further work and the potential for improvement but it is far from sufficient to determine that one method is inherently more accurate in general (research question 2). Furthermore, as previously stated the additional errors likely due to inaccuracies in retrieved parameters used to apply each method needs to be considered. Summary statements about reduction in errors etc all need to consider the realism of the weighting of the cases and angles in the derivation of these median values. The range of values

investigated also needs to be stated in the conclusions to give context to statements such as 'mean absolute relative error decreases with increasing...'.

The discussion of research question 3 probably needs more detail in the conclusion for it to be clear here.

**Other points**

**Equations 11.** I think LHS should be (rvol)^3 as volume (4 $\pi r^3/3$) = mass / density.

**Equation 13.** $h$ is introduced here without explanation, is this $z$ in equation 10?

**Figure 1** is strangely placed and not properly introduced or fully discussed in the text what is the purpose of this figure at this point in the text.

**Figure 2**. The legend and much of the rest of the text is too small to read easily.

**Figure 3 and 4.** The panels are too small to see properly and could be better spaced to make better use of the space available.

**Figure 9** needs to be enlarged, particularly the top part. The red and blue lines are difficult to distinguish and it is not clear what they enclose may hatched regions of the +/- 10Wm-2 would be a viable alternative. The legend and text refer to top and bottom panels, but I think it should be left and right.

**Figure 10 and 11**, many of the whiskers go off the scale an unknown amount.

**Figure A1**, refers again to upper and lower panels when they are arranged left and right, these would benefit from being enlarged, also it doesn't appear to be referenced in the text anywhere.

**Throughout:** I'm not going to address minor grammatical issues until the major points are addressed except to note that I think it should be 'principal plane' not 'principle plane'.

---

## Author Comment (AC2)

Figure 1 Reff of MODIS frame

Figure 2 optical depth of MODIS frame

Fig. 3 Effective Radius with Nd=25

Fig. 4 Effective Radius with Nd=100

Fig. 5 Effective Radius with Nd=400

Fig. 6 Cloud optical depth with Nd=25

Fig. 7 Cloud optical depth Nd=100

Fig. 8 Cloud optical depth Nd=400

Fig. 9 cloud geometrical thickness Nd=25

Fig. 10 cloud geometrical thickness Nd=100

Fig. 11 cloud geometrical thickness Nd=400

Fig. 12 Global density plot of mean liquid cloud optical thickness against mean cloud effective radius for May 2007 based on ESA CCI dataset (<a href="https://climate.esa.int/en/projects/cloud/">https://climate.esa.int/en/projects/cloud/</a>) and the range of optical thicknesses and effective radii used for the semi-physical approach in red dots. And the range of cot and reff within the corresponding scene in orange.